# Energetic Polymer Possessing Furazan, 1,2,3-Triazole, and Nitramine Subunits

**DOI:** 10.3390/ijms24119645

**Published:** 2023-06-01

**Authors:** Pavel S. Gribov, Natalia N. Kondakova, Natalia N. Il’icheva, Evgenia R. Stepanova, Anatoly P. Denisyuk, Vladimir A. Sizov, Varvara D. Dotsenko, Dmitry B. Vinogradov, Pavel V. Bulatov, Valery P. Sinditskii, Kyrill Yu. Suponitsky, Mikhail M. Il’in, Mukhamed L. Keshtov, Aleksei B. Sheremetev

**Affiliations:** 1Zelinsky Institute of Organic Chemistry, Russian Academy of Sciences, 47 Leninsky Prosp., Moscow 119991, Russia; 2Mendeleev University of Chemical Technology, 9 Miusskaya pl., Moscow 125047, Russiailicheva.n.n@muctr.ru (N.N.I.); dotsenko.v.d@muctr.ru (V.D.D.);; 3Institute of Organoelement Compounds, Russian Academy of Sciences, Moscow 119991, Russia; 4Basic Department of Chemistry of Innovative Materials and Technologies, Plekhanov Russian University of Economics, 36 Stremyannyi Line, Moscow 117997, Russia

**Keywords:** diazide, nitramine, dialkyne, energetic polymer, energetic plasticizer, synthesis, thermal decomposition, combustion, burning rate

## Abstract

A [3 + 2] cycloaddition reaction using dialkyne and diazide comonomers, both bearing explosophoric groups, to synthesize energetic polymers containing furazan and 1,2,3-triazole ring as well as nitramine group in the polymer chain have been described. The developed solvent- and catalyst-free approach is methodologically simple and effective, the comonomers used are easily available, and the resulting polymer does not need any purification. All this makes it a promising tool for the synthesis of energetic polymers. The protocol was utilized to generate multigram quantities of the target polymer, which has been comprehensively investigated. The resulting polymer was fully characterized by spectral and physico-chemical methods. Compatibility with energetic plasticizers, thermochemical characteristics, and combustion features indicate the prospects of this polymer as a binder base for energetic materials. The polymer of this study surpasses the benchmark energetic polymer, nitrocellulose (NC), in a number of properties.

## 1. Introduction

The need to improve efforts to synthesize components of energetic materials is obvious since the requirements for such materials that are promising for future use are increasing from year to year. Explosives, propellants, and powders are usually multicomponent energetic materials. For instance, a multicomponent system of solid composite propellants (SCPs) usually includes an oxidizer (50–80%, oxygen-rich inorganic and/or organic compounds), combustible (up to 25%, high-calorific metals or their alloys, metal hydrides, boron and its derivatives, high nitrogen compounds, etc.), binder (10–25%, plasticized polymers), catalysts, and processing aids [1,2,3,4,5,6,7]. All these components, complementing each other, should provide the required effect—the development of the maximum thrust and the maximum increase in the speed of the aircraft in the process of propellant burning, ensuring the delivery of the certain mass to the maximum possible distance. The extraordinary promise of SCPs for space, military, and civilian applications have led to a rapid growth of innovative research in this area.

The improvement of binder components for powders and SCP is an important area of research. Back in the 1970s, it was shown that higher propellant efficiency indicators can be achieved if active oxygen is delivered not only by an oxidizer, but also contained in a binder, in a polymer, or plasticizer of which, for example, bears nitroxy groups [8,9,10]. The expediency of introducing high-enthalpy moieties into polymers, such as an azide group and other explosophoric units also became obvious [11,12,13,14,15,16,17].

Binders created on the basis of a polymer and a plasticizer enriched with explosophoric groups [18] gave a significant increase in the specific impulse for metallized SCPs. Such components are even more in demand for propellants based on solid high-enthalpy high-density, but oxygen-poor fillers, C_x_H_y_N_z_O_w_, for which the oxygen coefficient, α = w/(2x + 0.5y), is significantly less than 1, as well as for propellants containing aluminum hydride [19] or boron-containing fuel [20].

When screening new binder components for modern powders and SCPs, their characteristics such as enthalpy of formation, ΔH^0^_f_, coefficient of supply with oxidizing elements, α, hydrogen content, %H, and density, *ρ*, are being analyzed. Of course, these four characteristics are interdependent and, usually, the improvement (growth) of one of them leads to the deterioration (decrease) of the others. For example, it is difficult to increase the proportion of oxygen in an oxidant without reducing its formation enthalpy.

Academic and industrial laboratories around the world are trying to clarify which components, in what combination and ratio, should be used to create effective powders and SCPs, and they are getting closer, step by step, to the cherished goal. However, there are still many unresolved issues.

The creation of a binder with an optimal chemical composition, which, in combination with an oxidizer and other components, would be suitable for processing both an uncured propellant mass and a hardened charge with the desired physical and mechanical properties, is a very difficult task.

As is known, to ensure sufficient rheological properties of the uncured propellant mass, where solid fillers are aluminum, inorganic and organic oxidizers, or gas-generating components, and to impart satisfactory physical and mechanical properties of the cured charge, the required minimum volume content of the binder should be ca. 19% [21]. Evidently, such a percentage of the binder has a significant impact on the energy performance of SCP. Energetic polymers are making a huge impact on development of new energetic materials due to their potential to improve performance.

While a variety of high-enthalpy heterocyclic building blocks have been widely used in the creation of low-molecular-weight energetic compounds in recent years, only tetrazole building blocks are used relatively widely in the construction of energetic polymers [22,23,24,25]. The use of other azoles as structural subunits of the polymer chain in the chemistry of energetic compounds is still used much less frequently.

It is obvious that the introduction of high-nitrogen subunits and explosophoric groups into the polymer can have a profound effect on the energy and physicochemical properties; therefore, synthesis methods and structural modifications of the polymer must be carefully designed to be effective.

Furazans (1,2,5-oxadiazoles) occupy a privileged position in energy compound chemistry and appear in multiple blockbuster low-molecular-weight energetic materials [26,27,28,29]. However, methods for the synthesis of energetic polymers that would incorporate the furazan subunit are rare, and only a few examples of polymerization processes leading to them have been reported by us [30,31] and others [32] previously. Figure 1 shows known representative energetic polymers bearing the furazan subunit.

However, these methods are either difficult to scale, or the target polymers had an unsuitable complex of physicochemical properties. A detailed description of both the preparation and properties of polymers **1–3** has not been reported in the public literature.

At the same time, the incorporation of furazan ring in the plasticizer molecules had a significant impact on the development of energetic binders [33].

The introduction of both polar and nonpolar groups into the polymer favors the polymer–plasticizer and polymer–filler interactions, which are very important for regulating the properties of the solid composite propellant being created. Only with a wide range of polymers and plasticizers at hand, there is a chance to create a suitable binder.

Recently, while developing new effective components of energetic binders, we have reported a series of energetic plasticizers that combine azido and nitramino groups on a dialkyl ether backbone, which have some specific properties realized due to the synergistic effect of the included functional moieties.

In our ongoing research focused on the development of new approaches leading to the formation of energetic molecules with combined functionality, here we report on the simple syntheses and characterization of energetic polyether, incorporated furazan and 1,2,3-triazol subunits, and a nitramine bridge. This structure will ensure the compatibility of the final polymer with other powder and SCP components, such as nitramines (for example, ammonium dinitramide (ADN), octahydro-1,3,5,7-tetranitro-1,3,5,7-tetrazocine (HMX), or hexanitrohexaazaisowurtzitane (CL-20)) or modern high-nitrogen energetic compounds.

Tasked with developing a practical route for the preparation of a densely functionalized energetic polymer, we were attracted by the possibility of constructing a polymer chain via a simple azide–alkyne cycloaddition using available comonomers. A survey of the literature revealed a number of examples of the use of this [3 + 2] cycloaddition reaction for the construction of energetic polymers [34,35,36]. While for the preparation of the target 1.2.3-triazole-based polymer, the initial monomer may contain both azido and alkyne moieties, the use of two comonomers, one of which contains two azido groups and the other two alkyne moieties, is more preferable for controlling the polymerization process.

To the best of our knowledge, there are no reports describing the use of any furazan derivatives in azide–alkyne cycloaddition polymerization. Moreover, we have not found in the literature any mention of the possibility of using diazido- and dialkynyl-comonomers, so that both contain explosophoric groups or units. Recently, the azide–alkyne cycloaddition is most often realized under copper catalysis, CuAAC reaction [37,38,39,40,41]. However, energetic polymers are used to create propellants that are designed for the combustion process. Copper derivatives have a great influence on the combustion process [42,43,44,45]. Since the copper content may be different in different batches of the polymer, this may lead to an uncontrolled change in the ballistic characteristics of the powders and SCPs containing this polymer. Therefore, the use of CuAAC in the production of polymers is undesirable in this case.

Herein, a copper-free thermally promoted azide–alkyne cycloaddition has been exploited to form a new energetic polymer. Such a protocol could be a useful tool for the simple development of promising energy binder components to meet diverse applications.

The resulting triazole–furazan-based nitraminopolymer has been studied both in terms of its energy properties and for its ability to plasticize, giving acceptable physicochemical properties to the binder.

## 2. Results and Discussion

### 2.1. Synthesis and Characterization of Monomers and Polymer

When an energetic compound can be synthesized simply and safely, and the starting materials are available and inexpensive, it is attractive for practical application. With this in mind, designing such bimodal monomers, wherein two molecules that serve two different functions can react with each other without the use of a catalyst and a solvent, is an ideal approach for producing energetic polymers. This protocol has a high atom economy, since all comonomers atoms are entered into the target polymer. 

As shown in Figure 2, the diazide monomer, 3,4-di(azidomethyl)furazan (**7**), was generated in three steps from commercially available dimethylglyoxime (**4**). Compounds **5** and **6** were prepared based on our previous report [46]. The nucleophilic azidation of (halogenomethyl)furazans is well known [47,48], and thus, diazide **7** was formed in quantitative yield by using three equivalent of NaN_3_ in acetone with **6** at room temperature.

Our second task was to devise a practical approach toward a suitable alkynyl-monomer-bearing nitramine groups. To the best of our knowledge, there is only one report [49] on the synthesis of dipropargylic ether, the bridge group of which contained nitramino-units, namely 1,6-di(2-propyn-1-yloxy)-2,5-dinitro-2,5-diazahexane (**9**), which was selected as a starting point in the creation of an energetic polymer. However, neither synthetic details nor fully characterization for this dipropargylic ether are given in the patent [49]. We, therefore, have investigated the chlorine displacement in N,N′-bis(chloromethyl)nitramine **8** with propargyl alcohol. To establish optimal conditions, we used readily available 1,6-dichloro-2,5-dinitro-2,5-diazahexane (**8a**) [50] and propargyl alcohol as our exemplar reagent set (Figure 3).

After an extensive survey of ratio of reactants, solvents, catalysts, and temperature, there are several important features to note. First, a solvent and a catalyst are not required for the successful course of this reaction. Secondly, propargyl alcohol serves both as a solvent and a reagent when using an excess of 20 equivalents. After the reaction was completed, the excess of propargyl alcohol was removed by vacuum distillation; thus, this alcohol was recycled and re-used. Thirdly, during the reaction, dry nitrogen should be slowly passed through the reaction mixture to remove the HCl released. Finally, we were able to reduce the reaction time to 1 h (vs. 21 h in the patent [49]) and improve the yield (85% vs. 68% in the patent [49]) at room temperature. The best reaction conditions found in this screening resulted in a higher yield of product 9 using 20 eq of propargyl alcohol, a reaction temperature of 20 °C, and a short reaction time (1 h), and scales well in kilogram quantities.

Under similar conditions, monoether **10** was prepared in a quantitative yield.

Dipropargyl ether **9** is colorless solid, and monoether **10** is slightly yellow oil. These ethers can be stored at room temperature in the absence of light for a long time.

All compounds are well characterized by IR, ^1^H, ^13^C, and ^14^N NMR spectroscopic data as well as CHN analysis (see Appendix A). The structure of dipropargyl ether **9** was unambiguously confirmed by X-ray crystallography (Figure 1, Appendix A). Compound **9** crystallizes in space group P2_1_/c with a calculated density of 1.475 gcm^−3^ at 20 °C. An asymmetric unit cell contains half of the molecule which is located at the symmetry center. The nitro groups are oriented pseudo-trans relative to each other so that the pseudo-torsional angle N2-N1 … N1A-N2A is equal exactly to 180° (Figure 1). The propargyl groups are oriented “inside” the molecule and form an angle of 21.5(2)° with the planes of the corresponding nitramino groups. The C2 … N2 (C2A … N2A) interatomic separation (3.131(2)Å) is less than sum of van-der-Waals atomic radii (3.36Å [51]), which might imply π … π interaction between propargyl and nitramino groups.

To clarify this point, we optimized the isolated molecule at the M052X/def2tzvp approximation level (for more details, see Appendix A). The optimized molecular conformation is almost the same as observed experimentally. Topological analysis of the calculated electron density (using AIM theory [52,53]) revealed a critical bond point between C2 and N2 (C2A and N2A) atoms, which indicates an attractive noncovalent interaction between propargyl and nitramino groups which contribute to the stabilization of the observed conformation.

Using combustion calorimetry, it was found that diazide **7** and dipropargyl ether **9** have enthalpy of formation +781.6 and 43.5 kJ/mol [54], respectively. Previously, only calculated values of the enthalpy of formation for any propargyl ethers bearing explosophoric groups were published [35,55].

Before investigating the dipolar cycloaddition polymerization process, we needed to establish the effect of explosophoric units, the furazan ring, and the nitramine group on the reactivity of our monomers. Toward proving that the above energetic monomers are suitable for alkyne–azide cycloaddition chemistry, we examined the model reaction of diazide **7** with monopropargyl ether **10** under different conditions, expecting to obtain molecules containing 1,2,3-triazole rings, which is demonstrated in Figure 4. We started our study with copper(I)-catalysed cycloaddition of an alkyne and azide (CuAAC), as this is known selectively provides 1,4-substituted-1,2,3-triazoles [37,38,39,40,41]. The cycloaddition between diazide **7** and acetylene **10** (2 equiv.) in N,N-dimethylformamide (DMF) proceeded to completion within 6 h (^1^H NMR control) at room temperature upon the addition of a catalytic amount of CuSO_4_·5H_2_O (0.05 equiv.) and sodium ascorbate (0.1 equiv.) to give ditriazole **11a** as the only the product in good isolated yield (analytically pure sample, 69%).

In the absence of a catalyst, the reaction between diazide **7** and acetylene **10** in refluxing 1,2-dichloroethane (DCE, 83 °C) required 90 h to achieve full consumption of the reagents. The catalyst-free reaction gave an inseparable mixture of three regioisomeric di-1,2,3-triazoles **11a**, **11b,** and **11c** in a ratio of 5:6:2, as observed by the ^1^H NMR analysis of the crude product which displayed, for example, two distinctive peaks (8.26 and 8.24 ppm) for the 1,4-substituted isomers and one peak (7.83 ppm) for the 1,5-isomer (see, Figure 2), the chemical shifts being consistent with those reported [56]. The regiochemical assignments were based both on chemical shifts in the pure isomer **11a** and on a combination of HSQC spectroscopy and NMR correlation (for more details, see Appendix A).

Since the alkyne–azide cycloaddition chemistry is suitable for binding two small molecules both bearing explosophoric groups together to form a new compound with the required functionality, we tried to apply this approach in the field of energetic materials to synthesize a new energetic polymer.

With the bifunctional precursors **7** and **9** in hand, we proceeded to investigate conditions for the cycloaddition polymerization. In a typical experiment, diazide **7** undergo the reaction with one equivalent of diacetylene **9** only by heating at a certain temperature without using a solvent or catalyst (Figure 5).

The 1,2,3-triazole rings containing products were characterized by gel permeation chromatography (GPC) and ^1^H NMR methods without any workup or further purification. Performing the reaction at 30 °C for 60 h led to very low conversion of the starting monomers. At 60 °C, as shown in Table 1, the molecular weight of the product depends on the reaction time; however, a further increase in the duration of the reaction to 100 h did not lead to a noticeable increase in the molecular weight of the polymer **12**. The polydispersity (*Ð*_M_) values are in range from 1.42 to 2.44, increasing with the increase in molecular weight. Nevertheless, under these conditions, the average molecular weight remains low.

When the reaction was gradually heated to 80 °C and kept at this temperature for 37 h, a light yellow plastic product **12** in quantitative yield was obtained. Purification of the resulting polymer from the residues of low-molecular-weight impurities was achieved by double precipitation from the DMSO solution with methanol. A colorless powder of **12** was obtained in 89% yield. Relatively high molecular weight was achieved that possessed modest dispersity (Table 1, Entry 6, and Appendix A).

The FTIR spectrum of polymer **12** was compared with these of the initial comonomers **7** and **9** and the model compound **11**. Figure 3 clearly demonstrates the pertinent features and differences of these spectra. The spectrum of diazide **7** shows the presence of azido groups at ca. 2111 cm^−1^. In the FTIR spectrum of the diacetylene compound **9**, two characteristic absorption bands at ca. 3290 and 2117 cm^−1^ can be attributed to C≡CH groups, whereas absorption bands for nitramino groups were observed at ca. 1550 and 1350 cm^−1^. The spectrum of model compound **11** showed the indicative loss of the aforementioned acetylene and azide bands, as a result of complete 1,3-dipolar cycloaddition. A similar trend was observed in the FTIR spectrum of polymer **12**; here, however, there is a weak residual band at ca. 2115 due to the presence of terminal C≡CH and N_3_ groups.

Similar to the reaction that took place during the synthesis of model compound **11**, the catalyst-free polymerization leads to the formation of a product including regioisomerically substituted 1,2,3-triazole subunits. As outlined in Figure 4, the ^1^H NMR spectrum of the resulting polymer **12** clearly showed the asymmetry that is present in the product. In particular, this spectrum displayed the appearance of three separate signals characteristic of a proton in the triazole ring at *δ* ca. 7.7–8.4, confirming polymer formation. This is noteworthy; the ratio of isomeric 1,2,3-triazole subunits is almost the same as in the model compound (see, Figure 2). The residual terminal alkyne proton is also clearly visible in the spectrum at *δ* ca. 3.6 ppm (for comparison, see Appendix A, ^1^H MNR of **9**).

The ^13^C NMR spectrum (see, Appendix A) further confirms the conclusions made on the basis of FTIR and ^1^H NMR. This spectrum of polymer **12** displayed the disappearance of characteristic alkyne moiety signals at δ ca. 77.5 and 79.5. This gives reason to conclude that most of the azide and alkyne groups of monomers **7** and **9** have been successfully spent on the formation of a triazole ring. The ^13^C NMR spectrum confirmed the formation of the 1,2,3-triazole subunit with the appearance of the signal at 125.1 ppm (CH of the 1,5-isomer) and 133.9 ppm (CH of the 1,4-isomer) which is indicative of the heterocycle [56]. The ^1^H and ^13^C NMR shifts illustrated in Figure 2 and in Appendix A also support the assignment of proton and carbon shifts in the polymer **12**.

Taking into account the enthalpies of the formation (ΔH_f_^0^) for comonomers **7** and **9** (see above), as well as the fact that the cycloaddition reaction is very exothermic (between −209 and −272 kJ mol^−1^ [57]), the theoretical value for ΔH_f_^0^ of polymer **12** is +1.4 kJ g^−1^, indicating that **12** has a higher energy content than NG (ΔH_f_^0^ = −2.2 kJ g^−1^ [18]).

### 2.2. Thermal Analysis

According to differential scanning calorimetric (DSC) and thermogravimetric analysis (TGA) measurements (scanning at 10 °C min^−1^, Figure 5 and Table 2) compound **9** melted with sharp endothermic peaks at 83 °C and began to decompose at 237 °C. The total energy of decomposition was 2212 J g^−1^. The temperatures of the beginning of the mass loss and the intensive decomposition of **9** coincide, indicating that the mass loss is due to decomposition, not evaporation. When heated, diazide **7** evaporates almost completely to a temperature of 200 °C, i.e., before decomposition begins (Table 2).

For polymer **12**, DSC revealed a glass transition (*T*_g_) near 48 °C, as well as a small endothermic peak (~6 J g^−1^) at 58 °C, associated with melting. However, the most striking observation from the DSC for this polymer was that a sample with *M*_w_ = 6747 g mol^−1^ and a sample with a higher molecular weight *M*_w_ = 42,400 g mol^−1^ have the same glass transition temperature. It is important to point out that polymer **12** has a significantly lower glass transition temperature than the benchmark energetic polymer, nitrocellulose (**NC**) (at 11.9% N, *T*_g_ = 160 °C [58]). The TGA plot in Figure 5b shows a mass loss due to decomposition commencing at 248 °C, a significant mass loss occurring as the temperature rises to *T*_peak_, and a loss of 60% of the original mass at 360 °C. The data in Table 2 demonstrate that the thermal stability of polymer **12** is higher than that of **NC**.

DSC thermograms for the decomposition of polymer **12** at different heating rates (from 5 to 20 °C min^−1^) are depicted in Figure 6. As indicated in Table 3, the apparent decomposition rate constants (*k*) of **12** that were obtained by the Kissinger method [59] are in accordance to Equation *k* = 4.66*·*10^14^*·*exp(−21,155/T), with activation energy *E*_a_ = 175.9 kJ mol^−1^.

The polymer **12** contains three units that can provoke the initial stage of decomposition, namely the nitroamine group, triazole, and furazan rings. It is possible to estimate which unit is responsible for the thermal stability of the polymer based on the kinetic parameters of the decomposition of these units.

For comonomer **9** bearing nitramino groups, thermal decomposition under isothermal conditions in the temperature range from 170 to 210 °C was carried out in thin-walled glass manometers of the compensation type, the glass Bourdon gauge (see Appendix A). This nitramine decomposed in the melt with weak acceleration. The gas emission curves can be described by a first-order model with linear autocatalysis [60]. This makes it possible to obtain (i) the initial decomposition rate constant (*k*_1_), characterizing the destruction of the nitramino group, and (ii) the acceleration rate constant (*k*_2_), which is probably due to the interaction of decomposition products (nitrogen oxides) with the initial **9**. As can be seen from Figure 7 the decomposition rate constants of polymer **12** fall on a straight line, which is a continuation of the straight line describing the initial decomposition rate of nitramine **9** (*k*_1_).

The kinetic parameters of the decomposition of polymer **12** and 3,4-dimethylfurazan (DMF) [61] can be seen in Figure 7. As can be seen from the comparison, the decomposition rate of DMF is more than an order of magnitude slower than that of the polymer, which suggests that the furazan ring does not decompose under DSC conditions. At the same time, taking into account the published data [62,63], the thermal stability of 1,2,3-triazole ring is comparable to the stability of polymer **12**. Summarizing the above, it can be assumed that the initial stage of decomposition of polymer **12** includes the decomposition of both 1,2,3-triazole units and nitramine groups.

Comparison of kinetic data of decomposition of **NC** (13.9% N) [64] and polymer **12** (Figure 7), shows that the decomposition rate of **12** is almost three orders of magnitude less than **NC**.

### 2.3. Plastification

It is well known that the introduction of a plasticizer into a polymer leads to a decrease in the glass transition temperature and an improvement in technological characteristics. Figure 8 demonstrates the appearance of the resulting polymer **12** before and after thermostating at ~60 °C. At room temperature, the polymer is a solid and after warming up, it turns into a viscous resin.

Nitroglycerin (**NG**) is usually used as a traditional plasticizer for **NC**. Both of these components and compositions based on them were benchmarked in this study. Figure 9 shows a DSC plot for a mixture of **NC** (12.2% N) with **NG** in a 50:50 ratio. The glass transition temperature (*T*_g_) of this mixture is −52 °C, and there are no signs of phase decay.

Polymer **12** is well compatible with **NG**. In Figure 10, the DSC data indicate that *T*_g_ of the mixtures of polymer **12** with **NG** range from −52 to −12 °C depending on the ratio of components; thus, only at 70% **NG** in the mixture it is possible to reach the glass transition temperature inherent to the **NC**/**NG** mixture with a 50:50 ratio. The limit of compatibility of polymer **12** with diethylene glycol dinitrate (**DEGDN**) is 20% (**12**: **DEGDN** = 80:20, *T*_g_ = 2 °C, Figure 11).

Not only nitroesters such as **NG** and **DEGDN**, but also organic azides [2,65,66] such as **Z1**, **Z3**, **Z8,** and **Z12** (Figure 12) were used to plasticize polymer **12**. The use of azide plasticizers allows to reduce the combustion temperature of the binder including them in comparison with compositions based on nitro ester plasticizers.

Azide plasticizers **Z1**, **Z3**, and **Z8** are well compatible with polymer **12** in a 50:50 ratio (Figure 13). Only one relaxation transition associated with the glass transition process of the plasticized composition was recorded on all profiles. At the same time, polymer **12** is practically incompatible with azide **Z12**; after 5 days of thermostating at +70 °C of the polymer–plasticizer mixture, the mass of polymer **12** increased by only 3.67%.

For all of the above mixtures, only the relaxation transition caused by the glass transition process of plasticized compositions was recorded on thermograms. The results are summarized in Table 4. It should be noted that the glass transition temperatures of mixtures **12**/**Z1** (50:50) and **NC**/**NG** (50:50) are practically equal. The glass transition temperatures of mixtures **12**/**Z3** and **12**/**Z8** are 13 and 9 °C, respectively, higher than that of the **NC**/**NG** composition (50:50).

The polymer composition **12**/**NG** (60:40) is practically unable to flow at room temperature; the polymer **12**, containing up to 60% **NG**, is a highly viscous resin, and when the **NG** content increases to 70%, a viscous liquid is formed (Figure 14)

As expected, with an increase in the plasticizer content in the polymer, the viscosity of the composition decreases (Figure 15, Table 5). The viscosity of polymer **12** containing 60 and 70% **NG** practically does not depend on the shear stress, which indicates the destruction of intermolecular interactions between polymer chains. With a content of 40% plasticizer in the composition, the interaction between the polymer macromolecules is still preserved: the viscosity decreases with increasing shear stress.

### 2.4. Combustion

The burning rate (r_b_) and its dependence on pressure are key parameters that need to be available when creating new rocket propellants for various purposes [1,3,67,68]. The accumulation and analysis of experimental data on burning rates and temperature distribution [69] makes it possible to evaluate and, in the long term, influence the kinetics of decomposition of propellant components burning in the condensed phase.

Burning rate was determined in a constant pressure bomb with a volume of 2 l in a nitrogen atmosphere. The process was recorded using a pressure strain gauge and a digital oscilloscope. Both a high-molecular polymer (*M*_w_ = 42,400 g mol^−1^) and its low-molecular analogue (*M*_w_ = 6747 g mol^−1^) were used for the study. Charges for determining the burning rate were prepared by pouring the polymer **12** presoftened at 75–80 °C into transparent polycarbonate tubes with an inner diameter of 7.5 mm, an outer diameter of 11 mm. As comparison samples, charges from both comonomer **9** and **NC** were prepared by blind pressing at a pressure of 450 MPa.

The results of these combustion experiments are shown in Figure 16, and data on the laws of combustion are summarized in Table 6. A cursory inspection of the data presented in Figure 16 and Table 6 demonstrates that the burning rates of high-molecular polymer **12** and **NC** is almost the same. Polymer **12** is a non-volatile material with a relatively low combustion temperature and burning rate is undoubtedly determined by the heat release in the condensed phase.

At a pressure of 10MPa, the burning rate of the comonomer **9** turned out to be very low. This is typical for the combustion of aliphatic nitramines [70], which is explained by their high thermal stability. Low burning rates imply a long residence time of the compounds at boiling points and, accordingly, a large contribution of the heat release in the molten phase in the control of the combustion. In this case, pressure exponent in the burning rate law is determined by the ratio of the activation energy of the leading combustion reaction and the enthalpy of evaporation, which is responsible for the temperature change with increasing pressure in this zone. The higher burning rate of the polymer **12** compared to that of the comonomer **9** is undoubtedly due to the higher surface temperature of the polymer, which is most likely due to less evaporation. This assumption is supported by a lower burning rate of a low-molecular polymer (Figure 16 and Table 6).

From a practical point of view, reducing the pressure exponent n is highly desirable; the lower n, the more stable the operation of the rocket engine is in relation to pressure fluctuations. Table 6 shows clearly that plasticization of polymer **12** with **NG** reduces the pressure exponent n from 0.93 to 0.6. More significantly, the effect of **NG** on polymer **12** is significantly higher than on **NC** (Figure 17). The burning rate of **12**/**NG** is higher than that of **NC**/**NG**.

### 2.5. Gunpowder

The resulting polymer **12**, as a potential heat-resistant replacement for **NC**, for example, can be used in gunpowders. The gauge for rating the efficiency of gunpowders is powder force (*F*). *F* is the work that could be conducted by gaseous products formed at the consumption of one kilogram of gunpowder, freely expanding at atmospheric pressure as a result of their heating from 273 K to the combustion temperature (*T*_c_) [3,71,72]. It is important to note that the higher the combustion temperature, the more destructive the effect of outgoing hot gases on metal equipment, whether it is a gun barrel, a rocket nozzle, etc. Calculated parameters of **NC**/**NG** and **12**/**NG** compositions are summarized in Table 7.

As can be seen from Table 7, according to the thermodynamic calculation, the compositions **12**/**NG** (50/50) and **NC**/**NG** (70/30) are comparable in magnitude to *F*, whereas the combustion temperature of **12**/**NG** is better (755 K lower) compared **NC**/**NG**. A more favorable *T_c_* is provided by a higher nitrogen content in polymer **12** compared to **NC**.

## 3. Materials and Methods

IR spectra were recorded on a BrukerALPHA instrument in KBr pellets. The ^1^H, ^13^C, and ^14^N spectra were recorded on a Bruker AM-300 instrument (300.13, 75.47, and 21.69 MHz, respectively) at 299 K. The chemical shifts of ^1^H and ^13^C nuclei were reported relative to TMS, for ^14^N*–*relative to MeNO_2_, high-filed chemical shifts are given with a minus sign. Elemental analysis was performed on a PerkinElmer 2400 Series II instrument. Analytical TLC was performed using commercially pre-coated silica gel plates (Kieselgel 60 F_254_), and visualization was affected with short-wavelength UV light.

Gel permeation chromatography (GPC) measurements of molecular weights were performed in N-methylpyrrolidone using a Smartline HPLC Series KNAUER system and calibrated with polystyrene standards.

Thermal stability, relaxation, and phase transitions were studied by differential scanning calorimetry (DSC) using a Mettler Toledo DSC 822e module. Approximately 2 mg of compounds were weighed and placed into in a 40 µL aluminum crucible, sealed under air with the appropriate sample press, and then pierced with a needle to leave one hole of approximately 1 mm in diameter. The decomposition of a sample was carried out in a nitrogen atmosphere at a purge rate of 50 μL min^−1^. The temperature of the onset of intense decomposition (*T*_onset_) was taken as the temperature determining thermal stability. To study relaxation and phase transitions, the samples were uncontrollably cooled to −130 °C and then heated at a rate of 10 °C min^−1^. The temperature of the midpoint of the relaxation transition was taken as the glass transition temperature (*T*_g_). The glass transition process is accompanied by a change in the heat capacity of the sample ΔC_p_, which was also measured. The melting point (*T*_m_) of the individual compounds was determined as the temperature of the melting effect start point.

Thermal degradation was quantified using a TGA 822e Mettler Toledo (TA instruments). Thermograms were recorded under an N_2_ atmosphere at a heating rate of 10 K min^−1^ from 25 to 400 °C.

The burning rate was determined in a constant pressure device (Crawford bomb) with a volume of 2 L in a nitrogen atmosphere. Liquid compounds were mixed with 4% nitrocellulose (colloxylin 12% N). The dissolution of nitrocellulose was carried out from 1 to 2 h at 50–60 °C into transparent acrylic tubes of 7 mm i.d. The combustion process of the sample was recorded using a pressure strain gauge, which transmitted the signal to a digital oscilloscope. The start and end times of combustion were determined from oscillograms. The burning rate was calculated by dividing the sample height by the burning time and was related to the mean integral pressure during the experiment. The error in determining the burning rate does not exceed 3%.

Most of the reagents and starting materials were purchased from commercial sources and used without additional purification. The starting compound **6** [46] and **8a,b** [50] were synthesized by using previously reported procedures.

**Caution!** Although we have encountered no difficulties during preparation and handling of these compounds, they are potentially explosive energetic materials. Manipulations must be carried out by using appropriate standard safety precautions.

**3,4-Bis(azidomethyl)furazan** (**7**). To a solution of 3,4-bis(bromomethyl)furazan (**6**) (2.56 g, 10 mmol) in acetone (10 mL) was added NaN_3_ (1.95 g, 30 mmol) at room temperature to give a suspension. The reaction was monitored by TLC and completed in about 24 h. Solvent was removed under reduced pressure, and the residue was extracted with Et_2_O (3 × 15 mL). After evaporation of diethyl ether, the crude product was purified by flash chromatography (CCl_4_) on silica gel to afford **7** as a light yellow liquid (1.76 g, 98%). R_f_ = 0.7 (CH_2_Cl_2_). ^1^H NMR (DMSO-d_6_) δ 4.86 (s, 2H, CH_2_). ^13^C NMR (DMSO- d_6_) δ 42.2, 151.1. IR (KBr): 2967, 2940, 2107, 1443, 1339, 1279, 1185, 1027, 890, 791 cm^−1^. Anal. Calcd. for C_4_H_4_N_8_O (180.13): C, 26.67; H, 2.24; N, 62.21. Found: C, 26.72; H, 2.27; N, 62.13.

**1,6-Di(2-propyn-1-yloxy)-2,5-dinitro-2,5-diazahexane** (**9**). 1,6-Dichloro-2,5-dinitro-2,5-diazahexane (**8a**, 2.47 g, 10.0 mmol) was dissolved in dry propargyl alcohol (10.8 mL, 200.0 mmol) and stirred at room temperature, passing dry N_2_ through the solution for 1 h. The excess of propargyl alcohol was removed by vacuum distillation at 30 °C to afford a yellow residue. After addition of H_2_O (30 mL), the mixture was stirred for 10 min. The flaxen precipitate was filtered, washed with water (3 × 30 mL), and dried. The crude product was recrystallized from CHCl_3_ to give **9** (2.43 g, 85%) as a white solid, mp 83-84 °C. ^1^H NMR (DMSO-d_6_) δ 3.47 (s, 1H, CH), 4.08 (s, 2H, CH_2_CH_2_), 4.25 (d, 2H, J = 1.9 Hz, CH_2_C), 5.21 (s, 2H, NCH_2_O). ^13^C NMR (DMSO-d_6_) δ 47.9, 56.5, 77.6, 78.7, 79.5. ^14^N NMR (DMSO-d_6_) δ −29.9 (NO_2_). IR (KBr): 3293, 2928, 2120, 1527, 1438, 1290, 1269, 1113, 1069, 1026, 981, 891, 891, 844, 663, 606 cm^−1^. Anal. calcd. for C_10_H_14_N_4_O_6_ (286.24): C 41.96, H 4.93, N 19.57. Found: C 42.02, H 4.96, N 19.61.

**1-(2-Propyn-1-yloxy)-2-nitro-2-azapropane** (**10**). The procedure is the same as for **9**. After evaporation of an excess of propargyl alcohol, the crude product was purified by flash chromatography (CH_2_Cl_2_, *R*_f_ = 0.60) on silica gel. The title compound **10** (97%) is a light yellow liquid; ^1^H NMR (DMSO-d_6_) δ 3.35 (s, 3H, CH_3_), 3.49 (t, 1H, J = 2.2 Hz, CH), 4.26 (d, 2H, J = 2.2 Hz, CH_2_C), 5.22 (s, 2H, NCH_2_O). ^13^C NMR (DMSO-d_6_) δ 37.9, 56.5, 77.5, 79.3, 79.6. ^14^N NMR (DMSO-d_6_) δ −28.2 (NO_2_). IR (KBr): 3288, 2955, 2930, 2864, 2119, 1532, 1473, 1435, 1299, 1250, 1080, 1048, 995, 978 cm^−1^. Anal. calcd. for C_5_H_8_N_2_O_3_ (144.13): C 41.67, H 5.59, N 19.44. Found: C 41.71, H 5.61, N 19.39.

**3,4-Bis((4-(2-methoxy-2-nitro-2-azapropane)-1H-1,2,3-triazol-1-yl)methyl)furazan** (**11a**). To a solution of compound **7** (0.18 g, 1 mmol) and **10** (0.288 g, 2 mmol) in DMF (5 mL) was added CuSO_4_**·**5H_2_O (0.013 g, 0.05 mmol) followed by sodium ascorbate (0.019 g, 0.1 mmol) under argon. The reaction mixture was stirred at room temperature for 6 h and then diluted with H_2_O (50 mL) and extracted using CH_2_Cl_2_ (5 × 20 mL). Combined organic layers were filtered through Celite, dried over MgSO_4_, filtered, concentrated on a rotary evaporator, and recrystallized from dichloroethane to give colorless solid (0.323 g, 69%), mp 117–119 °C. *R*_f_ = 0.25 (MeCN:CCl_4_ 1:3). ^1^H NMR (DMSO-d_6_) δ 3.37 (s, 3H, CH_3_), 4.68 (s, 2H, OCH_2_), 5.24 (s, 2H, NCH_2_O), 6.01 (s, 2H, Het-CH_2_-Het’), 8.26 (s, 1H, CH). ^13^C NMR (DMSO-d_6_) δ 37.4, 41.5, 61.5, 79.2, 124.6, 143.2, 150.3. IR (KBr): 3141, 3101, 2975, 2953, 1537, 1519, 1475, 1430, 1318, 1293, 1252, 1311, 1106, 1073, 1056, 1000, 938, 800 cm^−1^. Anal. Calcd. for C_14_H_20_N_12_O_7_ (468.39): C, 35.90; H, 4.30; N, 35.89. Found: C, 36.01; H, 4.24; N, 35.77.

**Mixture of isomers 11a–11c** (Catalyst free conditions). To a solution of compound **7** (0.18 g, 1 mmol) and **10** (0.288 g, 2 mmol) in DCE (5 mL) was heated under reflux until TLC analysis (3:2 hexanes–EtOAc) indicated complete consumption of starting materials (~90 h). Then DCE was evaporated in vacuo to dryness. Flash chromatography (CHCl_3_) furnished the product as an inseparable mixture of isomers **11a–11c** in 60–65% yield. The ratio of isomers was determined by ^1^H NMR spectroscopy of the crude and the purified isomer mixture (the ratio is invariable).

**Isomer 11a**: ^1^H NMR (DMSO-d_6_) δ 3.37 (s, 3H, CH_3_), 4.69 (s, 2H, OCH_2_), 5.25 (s, 2H, NCH_2_O), 6.01 (s, 2H, Het^1,4^-CH_2_-Het’), 8.26 (s, 1H, CH). ^13^C NMR (DMSO-d_6_) δ 37.9, 42.1, 62.1, 79.8, 125.2, 143.7, 150.8. IR (KBr): 3141, 3101, 2975, 2953, 1537, 1519, 1475, 1430, 1318, 1293, 1252, 1311, 1106, 1073, 1056, 1000, 938, 800 cm^−1^.

**Isomer 11b**: ^1^H NMR (DMSO-d_6_) δ 3.37 (s, 3H, CH_3_), 4.81 (s, 2H, OCH_2_), 5.18 (s, 2H, NCH_2_O), 5.94 (s, 2H, Het^1,5^-CH_2_-Het’), 7.83 (s, 1H, CH). ^13^C NMR (DMSO-d_6_) δ 38.0, 40.8, 58.7, 79.8, 134.1, 143.7, 150.6.

**Isomer 11c**: ^1^H NMR (DMSO-d_6_) δ 3.37 (s, 3H, CH_3_), 4.69 (s, 2H, OCH_2_), 4.82 (s, 2H, OCH_2_), 5.18 (s, 2H, NCH_2_O), 5.25 (s, 2H, NCH_2_O), 5.97 (s, 2H, Het^1,5^-CH_2_-Het’), 5.98 (s, 2H, Het^1,4^-CH_2_-Het’), 7.83 (s, 1H, CH), 8.24 (s, 1H, CH). ^13^C NMR (DMSO-d_6_) δ 37.9, 40.8, 42.2, 58.7, 62.1, 79.8, 125.2, 134.0, 134.1, 143.7, 150.6, 150.8.

**Polymer 12**. To a 50 mL beaker immersed in a water bath was added diazide **7** (10 g, 55.56 mmol). After the diazide was heated to 45–50 °C, diacetylene **9** (15.9 g, 55.56 mmol) was added portion-wise maintaining 45–50 °C. The oily reaction mixture was then stirred at 50 °C overnight, after which it was transferred to a Teflon mold and kept at 80 °C in a thermostat at reduced pressure for 37 h. The reaction mixture was allowed to cool to ambient temperature. The resulting target polymer **12**, which is a light yellow transparent plastic, was used in subsequent investigations without additional purification. An analytical sample was prepared by double precipitation from the DMSO solution with methanol, after which the precipitate was filtered, washed with methanol, and dried in vacuum for a day at 60 °C, which gave a colorless powder (89%). ^1^H NMR (DMSO-d_6_) δ 3.52, 4.09 (s, 2H, 1,4-triazole–N(NO)_2_CH_2_CH_2_), 4.11 (s, 2H, N(NO)_2_CH_2_CH_2_–Het^1,5^), 4.27 (s, 2H, CH_2_CCH), 4.68 (s, 2H, OCH_2_– Het^1,5^), 4.81 (s, 2H, OCH_2_–Het^1,4^), 5.19 (s, 2H, N(NO)_2_CH_2_OCH_2_–Het^1,5^), 5.24 (s, 2H, N(NO)_2_CH_2_OCH_2_–Het^1,4^), 5.93 (s, 2H, Het^1,5^–CH_2_–furazan–CH_2_–Het^1,5^), 5.96 (s, 2H, Het^1,4^–CH_2_–furazan–CH_2_–Het^1,5^), 5.97 (s, 2H, Het^1,4^–CH_2_–furazan–CH_2_–Het^1,5^), 6.01 (s, 2H, Het^1,4^–CH_2_–furazan–CH_2_–Het^1,4^), 7.80 (s, 1H, Het^1,5^–CH), 8.22 (s, 1H, (Het^1,4^–CH)–CH_2_–furazan–CH_2_–(Het^1,5^–CH)), 8.24 (s, 1H, Het^1,4^–CH). ^13^C NMR (DMSO-d_6_) δ 42.5, 42.6, 48.4, 56.9, 59.1, 62.5, 78.1, 79.6, 125.6, 134.5, 144.1, 150.9, 151.3. IR (KBr): 3293, 3144, 2963, 2113, 1528, 1461, 1437, 1290, 1270, 1111, 1069, 1025, 981. Anal. Calcd. for C_14_H_18_N_12_O_7_ (466.39): C, 36.06; H, 3.89; N, 36.04. Found: C, 36.12; H, 3.92; N, 35.99.

## 4. Conclusions

It has been demonstrated that azide–alkyne cycloaddition copolymerization under solvent- and catalyst-free conditions can enable the synthesis of a hybrid polymer bearing nitramine, furazan, and 1,2,3-triazole subunits in the polymer backbone. The method described herein requires no purification steps of the synthesized polymer. Alkyne and azide groups can be easily introduced into the required difunctional readily available precursors, both incorporating explosophoric units. This ring-closure copolymerization allowed for efficient diazide and dialkyne comonomers coupling, yielding an energetic polymer with high yield and sufficiently narrow polydispersity. The formation of the copolymer was confirmed by NMR, FT-IR, and SEC. Due to the presence of the nitramine, furazan, and 1,2,3-triazole subunits within the polymer repeating unit, this polymer is better than the benchmark energetic polymer, nitrocellulose (**NC**), in terms of glass transition temperature, thermal stability, and enthalpy of formation, whereas their burning rates are close. The polymer of this study is well plasticized by energetic plasticizers, which opens up wide opportunities for its use in the compositions of gunpowders and rocket propellants.

To the best of our knowledge, this is the first example of combining nitramino-, furazan-, and triazole-subunits in a polymer architecture designed by the Hüisgen 1,3-dipolar cycloaddition reaction of diazide and dialkyne comonomers, both of which contain explosophoric groups. Therefore, it represents a new way to produce synthetic energetic polymers endowed with a number of useful properties. Due to the reliable, easy-to-use and reproducible result of this azide–alkyne cycloaddition copolymerization, as well as the ability to tune the structure of both initial comonomers, we are confident that this simple and modular approach will find utility in the creation of new nitrogen- and oxygen-rich polymers for various applications related to energy materials. Further investigations on the synthesis and study of the structure-properties relationships of new energetic polymers are continuing in our laboratory.

## Data Availability

Data available under request.

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
