# Peer review of "Energetic Polymer Possessing Furazan, 1,2,3-Triazole, and Nitramine Subunits"

_ijms, 2023, doi:10.3390/ijms24119645_

Round 1

Reviewer 1 Report

This paper deals with the synthesis of functional polymers. The authors developed energetic polymer containing high degree of nitrogen atoms such as furazan and triazole moieties. All the products are well-characterized. The contents in this paper are of basic interest. Therefore, I recommend publication in Int. J. Mol. Sci. as is. 

Author Response

We thank the reviewer for the high assessment of our work.

Reviewer 2 Report

The article is written thoroughly and clearly. I can only indicate minor inaccuracies that should be corrected.   1. When designing the tables, the authors made some of them in black and white, and some of them were colored. A single design should be used.   2. In table 2, NC should be highlighted in bold.   3. The section "Materials and methods" describes conventional scientific instruments and generally accepted methods. I believe that this information could be transferred to Supporting Information. However, this wish is offered at the discretion of the editorial board, since the similar content of "Materials and Methods" is available in many articles in IJMS.   4. The list of references for some references does not include DOI.   These minor remarks do not reduce the high level of this article. The article can be published after minor corrections are made.

Personally, I believe the manuscript is really good (highly innovative and rich in content). I initially planned to suggest it being accepted in its current state, but after a carefully checking, I thought it could be further improved if my previous comments were given to the authors.

A minor revision based on my previous comments should be enough.

The manuscript is well written in English.

Author Response

Dear Editor,

We thank the referees for their careful consideration of our manuscript, for the expressed wishes and for the critical comments. All necessary changes have been made in the text (the changes that I have made during the revision are highlighted with a yellow background). A point by point list of comments is given below:

Reviewer 1

This reviewer did not find any topics in our manuscript that need correction.

Reviewer 2

Comment 1: When designing the tables, the authors made some of them in black and white, and some of them were colored. A single design should be used.

Response: This is corrected. Now all tables are decorated in color.

Comment 2: In table 2, NC should be highlighted in bold.

Response: This is corrected.

Comment 3: The section "Materials and methods" describes conventional scientific instruments and generally accepted methods. I believe that this information could be transferred to Supporting Information. However, this wish is offered at the discretion of the editorial board, since the similar content of "Materials and Methods" is available in many articles in IJMS.

Response: This section is designed by us in accordance with the rules of the Journal. If in the future the editorial board deems it appropriate to move this section to the Supporting Information, we will be able to do so.

Comment 4: The list of references for some references does not include DOI.

Response: Unfortunately, not all articles have DOI. Where possible, a DOI has been added.

A number of minor edits have also been made to the manuscript, which are highlighted with a marker.

On behalf of authors,

Sincerely yours,

Aleksei Sheremetev